# Infection with *Mycobacterium tuberculosis* alters the antibody response to HIV-1

Marius Zeeb[1,2]*, Katharina Kusejko[1,2], Sonja Hartnack[3], Chloé Pasin[1,2], Irene A. Abela[1,2], Peter Rusert[2], Thomas Liechti[2], Claus Kadelka[2], Julia Notter[4], Anna Eichenberger[5], Matthias Hoffmann[6], Hans H. Hirsch[7,8,9], Alexandra Calmy[10,11], Matthias Cavassini[12], Niklaus D. Labhardt[13,14], Enos Bernasconi[11,15,16], Huldrych F. Günthard[1,2], Roger D. Kouyos[1,2], Alexandra Trkola[2◉], Johannes Nemeth[1◉], and the Swiss HIV Cohort Study[¶]

1 Department of Infectious Diseases and Hospital Epidemiology, University Hospital Zurich, Zurich, Switzerland, 2 Institute of Medical Virology, University of Zurich, Zurich, Switzerland, 3 Section of Epidemiology, Vetsuisse Faculty, University of Zurich, Zurich, Switzerland, 4 Division of Infectious Diseases and Hospital Epidemiology, Cantonal Hospital Sankt Gallen, St. Gallen, Switzerland, 5 Department of Infectious Diseases, Inselspital University Hospital Bern, Bern, Switzerland, 6 Clinic for Infectious Diseases, Olten Cantonal Hospital, Olten, Switzerland, 7 Division of Infectious Diseases and Hospital Epidemiology, University Hospital Basel, Basel, Switzerland, 8 Department of Medical Biology, Faculty of Health Sciences, UiT The Arctic University of Tromsø, Tromsø, Norway, 9 Department Biomedicine, Transplantation and Clinical Virology, University of Basel, Basel, Switzerland, 10 HIV/AIDS Unit, Division of Infectious Diseases, University Hospital Geneva, Geneva, Switzerland, 11 Faculty of Medicine, University of Geneva, Geneva, Switzerland, 12 Division of Infectious Diseases, University Hospital of Lausanne, Lausanne, Switzerland, 13 Division Clinical Epidemiology, Department of Clinical Research, University Hospital Basel, Basel, Switzerland, 14 University of Basel, Basel, Switzerland, 15 Division of Infectious Diseases, Repubblica e Cantone Ticino Ente Ospedaliero Cantonale, Lugano, Switzerland, 16 University of Southern Switzerland, Lugano, Switzerland

¶ Membership of Swiss HIV Cohort Study is provided in Supporting Information file S1 Acknowledgements
◉ These authors contributed equally to this work.
* marius.zeeb@usz.ch

## Abstract

### Background:

Co-infection with *Mycobacterium tuberculosis* (MTB) differentially modulates untreated HIV-1 infection, with asymptomatic MTB reducing HIV-1 viremia and opportunistic infections and active tuberculosis (TB) accelerating AIDS progression. Here, we investigate antibody (Ab) responses to HIV-1 in people with HIV (PWH) without MTB, with asymptomatic MTB, and with later progression to active TB to elucidate MTB-associated effects on HIV-1 immune control.

### Methods:

Using the Swiss HIV Cohort Study (SHCS), we conducted a retrospective study that included 2,840 PWH with data on MTB status and HIV-1-specific plasma binding-/neutralizing-responses. We evaluated associations between MTB status and binding-/neutralizing-responses while adjusting for key disease and demographic parameters.

**Data availability statement:** The individual level datasets generated or analyzed during the current study do not fulfill the requirements for open data access: 1) The SHCS informed consent states that sharing data outside the SHCS network is only permitted for specific studies on HIV infection and its complications, and to researchers who have signed an agreement detailing the use of the data and biological samples; and 2) the data is too dense and comprehensive to preserve patient privacy in persons living with HIV. According to the Swiss law, data cannot be shared if data subjects have not agreed or data is too sensitive to share. Investigators with a request for selected data should send a proposal to the respective SHCS address (www.shcs.ch/contact). The provision of data will be considered by the Scientific Board of the SHCS and the study team and is subject to Swiss legal and ethical regulations, and is outlined in a material and data transfer agreement.

**Funding:** This work was funded within the framework of the Swiss HIV Cohort study (SHCS) supported by the Swiss National Science Foundation (grant number 324730_207957) (https://www.snf.ch/en) and by the Swiss HIV Cohort Study research foundation (https://shcsfoundation.ch/) (Project No. 920). The funders had no role in study design, data collection and analysis, decision to publish, or preparation of the manuscript.

**Competing interests:** I have read the journal's policy and the authors of this manuscript have the following competing interests: K.K. has received research grants unrelated to this work from the Swiss National Science Foundation. I.A.A. received research grants from the Swiss HIV Cohort Study, Gilead and travel expenses from Gilead for research unrelated to this work. J.No. received research grants unrelated to this work from the Swiss HIV Cohort Study and the cantonal hospital St. Gallen, paid to her institution, and travel expenses from Gilead unrelated to this work. M.C. received research grants unrelated to this work from Gilead, ViiV and MSD, paid to his institution; payment for expert testimony unrelated to his work from Gilead, ViiV and MSD, paid to his institution; and travel expenses unrelated to this work from Gilead, paid to his institution. A.E. received honoraria for presentations unrelated to this work from

## Results:

Among the included 2,840 PWH, 263 PWH had asymptomatic MTB based on either a positive TST-/IGRA-test at the baseline (time of HIV-1 Ab measurement) or on later progression to active TB. Compared to PWH without MTB infection, PWH with asymptomatic MTB infection showed reduced HIV-1 Ab levels, both for Env binding (e.g., IgG1 BG505 trimer antigen, $p = 0.024$) and neutralization of a diverse panel of HIV-1 viruses ($p = 0.012$). Conversely, PWH ($n = 32$) who later progressed to active TB (>180 days after baseline) demonstrated a significant shift towards IgG3 in their HIV-1 Ab repertoire ($p = 0.011$), detectable in median 3.8 years (IQR 2.4 – 8.7) before active TB onset.

## Conclusion:

Our data indicate that asymptomatic MTB infection and active TB exert profound heterologous effects on HIV-1 specific Ab development. These findings advance our understanding of host-pathogen dynamics and may have implications for new diagnostic approaches in predicting future active TB.

## Author summary

Active *Mycobacterium tuberculosis* (MTB) infection remains a leading cause of death among people with HIV-1 (PWH). However, the majority of PWH infected with MTB do not progress to active tuberculosis. Interestingly, MTB infection has also been associated with beneficial clinical outcomes, such as lower HIV RNA levels and a reduced incidence of opportunistic infections. In light of these findings, we here investigated whether MTB infection modulates the adaptive immune response against HIV-1. To do so, we analysed a well-defined subpopulation of 2,840 PWH from the Swiss HIV Cohort Study, for whom detailed immunological profiles of HIV-1 antibody responses were available. We found that MTB infection was associated with both reduced HIV-1 neutralization capacity and lower HIV-1 IgG1 antibody binding reactivity. This is most likely a consequence of decreased HIV RNA levels, potentially due to MTB-induced modulation of the innate immune system, leading to reduced HIV-1 antigen presence and, consequently, weaker HIV-1 antibody responses. In contrast, we observed that higher HIV-1 IgG3 antibody reactivity was predictive for progression to active MTB disease. Our findings provide new insights into the immunological interplay between HIV-1/MTB co-infection and may inform development of novel diagnostic approaches.

## Introduction

*Mycobacterium tuberculosis* (MTB) remains a major global health threat, with 1.3 million deaths attributed to MTB infection in 2022 [1]. MTB is highly infectious, but the

Bavarian Nordic, paid to her institution; and honoraria for advisory board consultations unrelated to this work from ViiV and Gilead, paid to her institution. E.B. received research grants unrelated to this work from MSD, paid to his institution; consulting fees unrelated to this work from Moderna, paid to his institution; honoraria for presentations unrelated to this work from Pfizer, paid to his institution; travel expenses unrelated to this work from ViiV, MSD, Gilead and Pfizer, paid to his institution; and honoraria for data safety monitoring board or advisory board consultations unrelated to this work from ViiV, MSD, Pfizer, Gilead, Moderna, AstraZeneca, AbbVie and Ely Lilly, paid to his institution. H.F.G. has received research grants unrelated to this work from the Swiss National Science Foundation, Swiss HIV Cohort Study, Yvonne Jacob Foundation, NIH, Gilead, ViiV and Bill and Melinda Gates foundation, paid to his institution; personal honoraria for data safety monitoring board or advisory board consultations unrelated to this work from Merck, ViiV healthcare, Gilead Sciences, Janssen, Johnson and Johnson, Novartis and GSK. R.D.K. received research grants unrelated to this work from Gilead and NIH, paid to his institution. All other authors have declared that no competing interests exist.

vast majority of individuals exposed to MTB do not develop active TB disease and effectively clear or contain their infection [2–6].

In people with HIV (PWH) who do not receive antiretroviral therapy (ART), MTB has long been recognized as a risk factor for progression to AIDS [7,8]. Although the number of HIV-1-associated TB cases has been decreasing due to the wider use of antiretroviral treatment regimens [1], HIV-1 infection continues to be an important risk factor for the development of active TB [3,5,6,8–20]. While the detrimental effect of active TB on HIV-1 progression is evident, asymptomatic MTB infection surprisingly correlates with improved HIV-1 control and reduced opportunistic infections [21]. Contrasting this, asymptomatic MTB infection has been shown to pose an increased risk for non-communicable diseases [22,23] suggesting that MTB infection causes perturbations of the immune system that differentially influence other diseases.

The immune factors that lead to natural control of MTB infection are not fully understood. Efforts to elucidate key components of adaptive and innate immunity to MTB are underway and aimed at using knowledge of the natural control of MTB for prevention and therapy [4–7]. Next to documented effects of MTB infection on T cell responses and innate immunity, in particular natural killer cell responses [24–27], shifts in MTB antibody (Ab) responses have been suggested to potentially impact the outcome of MTB infection [28–30]. There is increasing evidence that the Ab response to MTB differs between asymptomatic MTB infection and active TB [31–33]. In particular, functional profiles of the immunoglobulins' fragment crystallizable (Fc) and Ab glycosylation patterns vary, with Abs during asymptomatic MTB infection showing enhanced phagolysosomal maturation, inflammasome activation, and macrophage killing of intracellular MTB [31,34]. Notably, a distinct IgG class shift has been documented between asymptomatic MTB infection and active TB, with a decrease in TB specific IgG3 levels in active TB and recurrent TB, while IgG1 levels increase in active TB, suggesting a protective role of IgG3 against active TB [35,36].

Untreated HIV-1 infection is known to cause a severe dysregulation of the B cell response, including abnormal activation of B cells leading to hypergammaglobulinemia, B cell exhaustion and reduced capacity to create de novo antigen responses [37–39]. IgG3 responses to HIV-1 peak in acute infection and typically decline to modest levels after 4 weeks of infection [40], while IgG1 specific to HIV-1 dominates [41].

Considering that both HIV-1 and MTB infection show distinct patterns of Ab responses across disease stages we here used the database of the Swiss HIV Cohort Study (SHCS) [42] to investigate the impact of MTB infection on HIV-specific Ab responses to gain insights into if and how MTB co-infection affects humoral immunity against HIV-1.

## Materials and methods

### Ethic committee approval and informed consent

The SHCS was approved by the local ethical committees of the participating centres:

Ethikkommission beider Basel ("Die Ethikkommission beider Basel hat die Dokumente zur Studie zustimmend zur Kenntnis genommen und genehmigt."); Kantonale

Ethikkommission Bern (21/88); Comité départemental d'éthique des spécialités médicales et de médecine communautaire et de premier recours, Hôpitaux Universitaires de Genève (01–142); Commission cantonale d'éthique de la recherche sur l'être humain, Canton de Vaud (131/01); Comitato etico cantonale, Repubblica e Cantone Ticino (CE 813); Ethikkommission des Kantons St. Gallen (EKSG 12/003); Kantonale Ethikkommission Zürich (KEK-ZH-NR: EK-793), and written informed consent was obtained from all participants to the anonymized use of their data and specimen via the SHCS database and biobank for retrospective studies.

## Study population

This retrospective study (Fig 1) utilized the database of the SHCS [42] which includes information on MTB status, and published HIV-1 plasma Ab data (neutralizing and binding Abs) from SHCS participants [43–45]. The latter HIV-1 Ab screening timepoint is the point of reference (baseline) in the present study. SHCS participants with untreated HIV-1 infection and available MTB status (see below) at baseline were included. The SHCS is a prospective, nationwide, longitudinal, observation, clinic-based cohort enrolling PWH in Switzerland since 1988 and registered under the Swiss National Science longitudinal platform program (https://www.snf.ch/en/YA2SxeDV03G25gyJ/funding/programmes/longitudinal-studies#Currently%20supported%20longitudinal%20studies).

## Demographic characteristics and HIV-1 disease specific parameters

Demographic characteristics and HIV-1 disease specific parameters at baseline used in this study were available through the SHCS database and included MTB status, ethnicity (Black, Other, reference: White)), age (age in decades), HIV-1

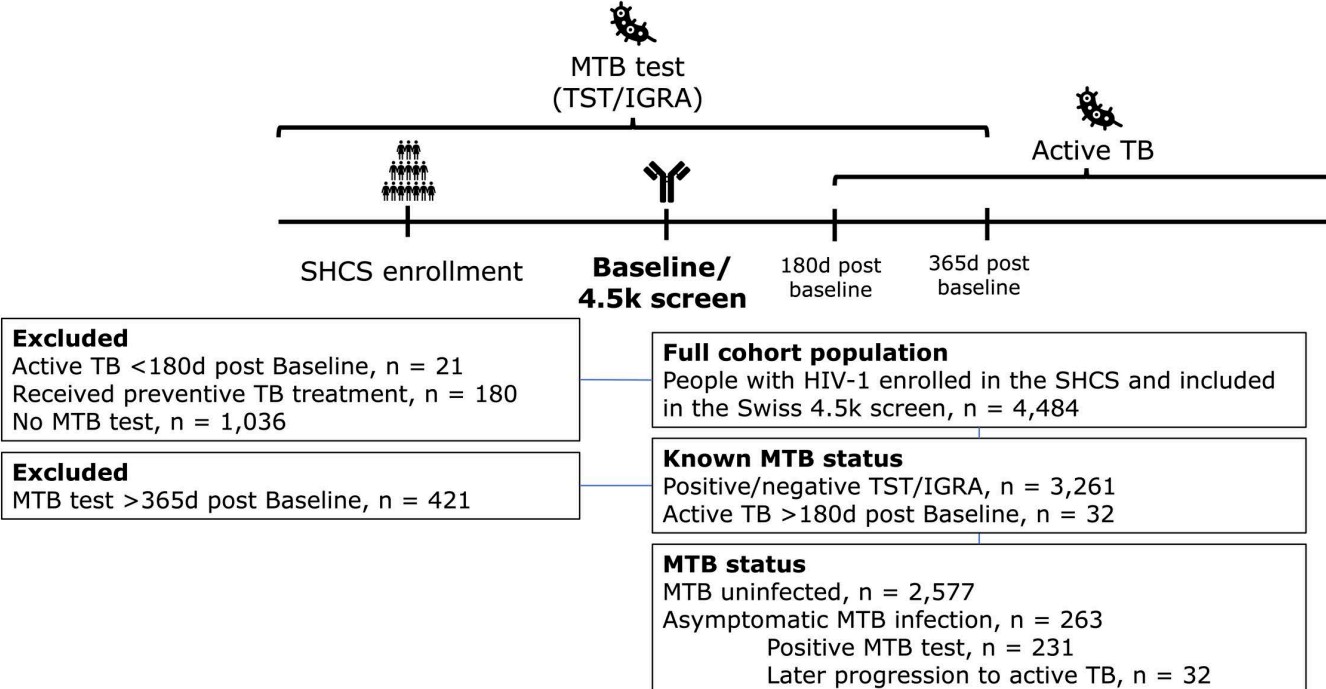

Fig 1. Study population selection. People with HIV from the SHCS and Swiss 4.5k screen included and excluded depended on their MTB status. Abbreviations: IGRA (Interferon gamma release assay), MTB (Mycobacterium tuberculosis), SHCS (Swiss HIV Cohort Study), TB (Tuberculosis), TST (tuberculin skin test). Figure was created with Microsoft PowerPoint and icons from UXWing (https://uxwing.com/antibodies-icon/; https://uxwing.com/bacteria-icon/).

transmission mode (heterosexual (HET), intravenous drug use (IDU), Other, reference: Men Who Have Sex with Men (MSM)), sex (female, reference: male), CD4 + T cell count (square root transformed CD4 T cell count), HIV-1 RNA viral load (log10 transformed HIV-1 RNA viral load), HIV-1 diversity (proportion of ambiguous nucleotides in the HIV-1 *pol* gene derived from Sanger sequencing) [46], time since infection (cumulative time off ART since HIV-1 infection in decades), and HIV-1 subtype (non-B, reference: B).

## Clinical phenotype

We focused our analyses on PWH with an inferable MTB infection status at baseline, i.e., the timepoint of HIV-1 Ab measurement. We defined MTB status as follows: i) MTB uninfected, defined through presence of a negative tuberculin skin test (TST) or interferon gamma release assay (IGRA) any time before or 1 year post baseline. ii) Asymptomatic MTB infection, defined either through a positive TST or IGRA any time before or 1 year post baseline, or as progression to active TB > 180 days post baseline (Fig 1). PWH with active TB at ≤180 days of baseline were excluded. PWH who received TB preventive treatment (TPT) were excluded.

## HIV-1 binding and neutralizing antibody measurements

HIV-1 neutralizing and binding Ab data of SHCS participants were collected in the frame of the Swiss 4.5K Screen [43–45]. All included plasma samples were retrospectively taken from the biobank of the Swiss HIV Cohort Study, from a sampling timepoint off ART and with a detectable viral load. People had to have an untreated HIV infection for one year or longer since the time of infection. ART treatment interruptions counted towards the one year. In case of treatment interruption, the sample had to be a least 90 days post interruption [45]. We included plasma neutralization data against a multi-clade 8-virus pseudovirus panel published in Rusert et al. [45]. Neutralization was determined from observed plasma neutralization against eight HIV-1 strains from five subtypes, ranging from 0 to 100% (S1 Fig and S1 Table). For each strain a score was calculated, 3 when neutralization >80%, 2 when neutralization 50% to 80%, 1 when neutralization 20% to 50%, and 0 when neutralization <20%. The sum of the individual virus scores were then summarized as the neutralization score (ranging from 0 to maximum 24) [45].

Relative IgG1, IgG2, IgG3 plasma binding Ab measurements against 13 HIV antigens, 10 envelope (Env) antigens (trimer (BG505), gp140 (BG505), gp120 monomer (JR-FL), V3 (BG505, MN, JR-FL), resurfaced stabilized Core 3 (RSC3, RSCΔ), membrane proximal region (MPER) peptide (MPER-2/4, MPER-2/4/10), gp41 (gp41ΔMPER)), and two gag antigens (p17, p24), were derived from bead based binding Ab multiplex assay (BAMA) data published in Kadelka et al. [43] (S2 Table). IgG1 binding responses against p17, p24, gp120 JR-FL, and gp41ΔMPER were measured with 1:650 plasma dilution. All other binding responses were done with a 1:100 plasma dilution [43]. The mean fluorescent intensity (MFI) values were transformed to a relative binding ranging from 0 to 100, whereas 100 indicates strongest binding [43] (S2 Fig and S2 Table).

## Analysis of associations of the MTB status with neutralization score and HIV-1 binding Ab responses

The association between asymptomatic MTB infection and HIV-1 neutralization score was determined with a tobit regression model. Associations between asymptomatic MTB infection and HIV-1 Ab binding responses (individual antigens or combined IgG class) were determined with linear regression models. Later progression to active TB and associations with HIV-1 Ab binding responses (individual antigens or the average of responses within each IgG class) were determined using cox proportional hazard regression. All models were adjusted for demographic characteristics and HIV-1 disease specific parameters. We accounted for multiple testing with the Benjamini-Hochberg (BH) procedure [47].

## Sensitivity analyses

We restricted all analyses to those PWH who received an IGRA to account for the lower diagnostic precision of the TST.

We assessed if association patterns with asymptomatic MTB infection are also observed with seropositivity to other infections among PWH with known MTB status. We considered the serostatus of CMV, *Toxoplasma gondii*, and HCV (based on IgG antibody tests any time before or 1 year post baseline) and repeated the analyses on HIV-1 neutralization score and HIV-1 Ab binding.

We assessed if later progression to 19 other opportunistic infections and non-communicable conditions, besides active TB, is associated with IgG responses. We considered opportunistic infections and non-communicable conditions with n > 30 among the PWH with known MTB status: Oral candidiasis, Herpes zoster multidermatomal/relapse, Oral hairy leukoplakia, Esophageal candidiasis, Kaposi sarcoma, Non-Hodgkin's lymphoma, HIV-related encephalopathy, HIV-related thrombocytopenia, Pneumocystis pneumonia, Myocardial infarction, Bacterial pneumonia, Cerebral infarction, Cirrhosis, Diabetes mellitus, Herpes zoster monodermatomal, Osteoporosis, Pancreatitis, Deep vein thrombosis, and Pulmonary embolism.

## Results

### Establishing a study population of HIV-1/ MTB co-infection in the SHCS database

HIV-1 infection and MTB infection individually cause severe perturbations of B cell immunity but the effects of these dysregulations in the context of co-infection remain to be resolved [48–50]. Here we investigated HIV-1/MTB co-infection to explore if HIV-1 Ab responses are influenced by concurrent MTB infection.

We previously conducted the Swiss 4.5K Screen, a systematic, comprehensive analysis of HIV-1 neutralizing and binding Ab signatures in untreated HIV-1 infection among close to 4,500 PWH enrolled in the SHCS [43,45]. Among these, 2,840 PWH had data on their MTB status at the time of HIV-1 Ab screening and were included in the present study (Fig 1 and Table 1). We utilized a range of demographic data through the SHCS database (Table 1) and available neutralization data against eight virus strains and BAMA data for IgG1, IgG2, and IgG3 against 13 HIV-1 antigens (Env trimer (BG505), gp140 (BG505), gp120 monomer (JR-FL), V3 (BG505, MN, JR-FL), resurfaced stabilized Core 3 (RSC3, RSCΔ), membrane proximal region (MPER) peptide (MPER-2/4, MPER-2/4/10), gp41 (gp41ΔMPER), and gag (p24, p17)) at a fixed plasma dilution for our investigations.

For the current analysis, we first verified that HIV-1 Ab signatures in this sub-population of 2,840 PWH reflected the key associations, with demographic and diseases measures, we had observed in the full population [43,45] (S3 and S4 Figs; N = 2,840). We confirmed significant correlations between neutralization score and viral load, CD4, duration of HIV-1 infection, viral diversity, and black ethnicity (S3 Fig). Binding Ab analyses recorded strong positive correlations for IgG1 responses to Env antigens with duration of HIV-1 infection and viral diversity. Viral load was inversely correlated with anti-Gag IgG1 (S4 Fig).

We next stratified the 2,840 PWH according to their MTB status at the HIV-1 Ab plasma sampling timepoint baseline into PWH without MTB infection (N = 2,577 (90.7%)) and with asymptomatic MTB infection (N = 263 (9.3%)) (Figs 1 and S5 and Table 1). Comparing key demographic parameters and disease measures showed PWH with asymptomatic MTB infection had a significantly lower viral load (odds ratio (OR) = 0.72; 95% confidence interval (CI) = 0.61, 0.84), higher CD4 (OR = 1.93; CI = 1.44, 2.57), were less likely to be female (OR = 0.65; CI = 0.46, 0.91), and more likely of black ethnicity (OR = 3.49; CI = 2.24, 5.45) (Table 1 and S5 Fig). Of note, 32 PWH with asymptomatic MTB infection later progressed to active TB which we considered in subsequent analyses (Table 1).

### Effects of asymptomatic MTB infection on HIV-1 plasma neutralization activity

Viral load is an independent driver of broad neutralization activity [45]. As asymptomatic MTB infection is associated with lower viral loads [21], HIV-1/MTB co-infection may have several consequences for HIV-1 neutralizing activity. For PWH who have developed potent neutralization activity prior to MTB infection, both pre-existing neutralization activity and the effects of MTB may contribute to HIV-1 control. For PWH without neutralization breadth at the time of MTB infection, a

**Table 1. Study population characteristics stratified by MTB status.**

| Characteristics at plasma sampling | Overall N = 2,840 | MTB uninfected, N = 2,577 | Asymptomatic MTB infection, N = 263 | | |
|---|---|---|---|---|---|
| **Active TB progression** | | | Overall, N = 263 | Positive MTB test, N = 231 | active TB progression, N = 32 |
| Neutralization score[1] | 2 (0,4) | 2 (0,4) | 1 (0,3) | 1 (0,3) | 1 (0,4) |
| Ethnicity[2] | | | | | |
| White | 2,297 (81%) | 2,126 (82%) | 171 (65%) | 155 (67%) | 16 (50%) |
| Black | 344 (12%) | 273 (11%) | 71 (27%) | 58 (25%) | 13 (41%) |
| Other | 199 (7.0%) | 178 (6.9%) | 21 (8.0%) | 18 (7.8%) | 3 (9.4%) |
| Sex [female][2] | 892 (31%) | 797 (31%) | 95 (36%) | 80 (35%) | 15 (47%) |
| HIV transmission group[2] | | | | | |
| MSM | 1,150 (40%) | 1,072 (42%) | 78 (30%) | 72 (31%) | 6 (19%) |
| HET | 1,043 (37%) | 921 (36%) | 122 (46%) | 105 (45%) | 17 (53%) |
| IDU | 546 (19%) | 495 (19%) | 51 (19%) | 43 (19%) | 8 (25%) |
| Other | 101 (3.6%) | 89 (3.5%) | 12 (4.6%) | 11 (4.8%) | 1 (3.1%) |
| Age [years][1] | 37 (32,43) | 37 (32,44) | 36 (31,43) | 37 (32,43) | 33 (29,39) |
| Cumulative time off ART since HIV infection [years][1] | 4.2 (3.4,6.8) | 4.2 (3.4,6.8) | 4.2 (3.7,6.3) | 4.2 (3.7,6.6) | 4.3 (3.8,5.4) |
| Log10 HIV-1 viral load [cps/ml][1] | 4.29 (3.71,4.78) | 4.33 (3.75,4.80) | 3.99 (3.40,4.52) | 3.96 (3.39,4.47) | 4.29 (3.67,4.85) |
| CD4 T count [cells/µl][1] | 400 (300,541) | 393 (297,532) | 455 (360,610) | 460 (361,618) | 420 (349,568) |
| HIV-1 subtype [non-B][2] | 743 (26%) | 639 (25%) | 104 (40%) | 87 (38%) | 17 (53%) |
| HIV-1 diversity [%pol ambiguity][1] | 1.24 (0.56,2.23) | 1.24 (0.60,2.23) | 1.24 (0.55,2.26) | 1.13 (0.46,2.14) | 1.81 (1.09,2.95) |

[1]Median (IQR); [2] n (%).

reduction in viral load caused by MTB may make them less likely to develop bnAb activity. Following the calculation of the neutralization score by Rusert et al. [45] that ranges from 0 to 24, the current study population has a median neutralization score of 2 (Table 1). Notably, asymptomatic MTB-infected PWH showed reduced neutralization scores (p = 0.012, univariable Tobit model, Fig 2A). Neutralization against viral strains JR-FL, Q23, and CNE59 was significantly reduced in PWH with asymptomatic MTB compared to PWH without (S1 Fig).

We further assessed the neutralization score using multivariable Tobit models. Importantly, after adjusting for key demographic and HIV-1 disease parameters, asymptomatic MTB infection remained inversely correlated with the neutralization score (regression coefficient = -0.9; CI = -1.53, -0.27, Fig 2B). We conducted a sensitivity analysis including only PWH who underwent IGRA testing (demographic details in S3 Table). Within this subgroup, we found no significant association with HIV-1 neutralization capacity (S6A Fig). Nevertheless, the overall findings suggest that asymptomatic MTB infection is associated with reduced HIV-1 neutralization capacity.

## Effects of asymptomatic MTB infection on the HIV-1 binding antibody landscape

We next explored if MTB infection leads to shifts in the HIV-1 binding Ab response. We conducted two sets of multivariable linear regression analyses, controlling for demographic characteristics and HIV-1 disease specific parameters, including or excluding HIV-1 viral load. This allows to determine effects of MTB infection that may be linked to viral load reduction and such that occur independent of viral load (Fig 2C).

Controlling for all factors, including viral load, we observed trends of a reduced IgG reactivity in PWH with asymptomatic MTB. After adjusting for multiple testing this reduction was significant for IgG1 Abs against the antigens gp120 JR-FL, BG505 SOSIP trimer, RSC3Δ, and MPER-2/4/10 (p = 0.024 - 0.038). Excluding HIV-1 viral load did not result in changes

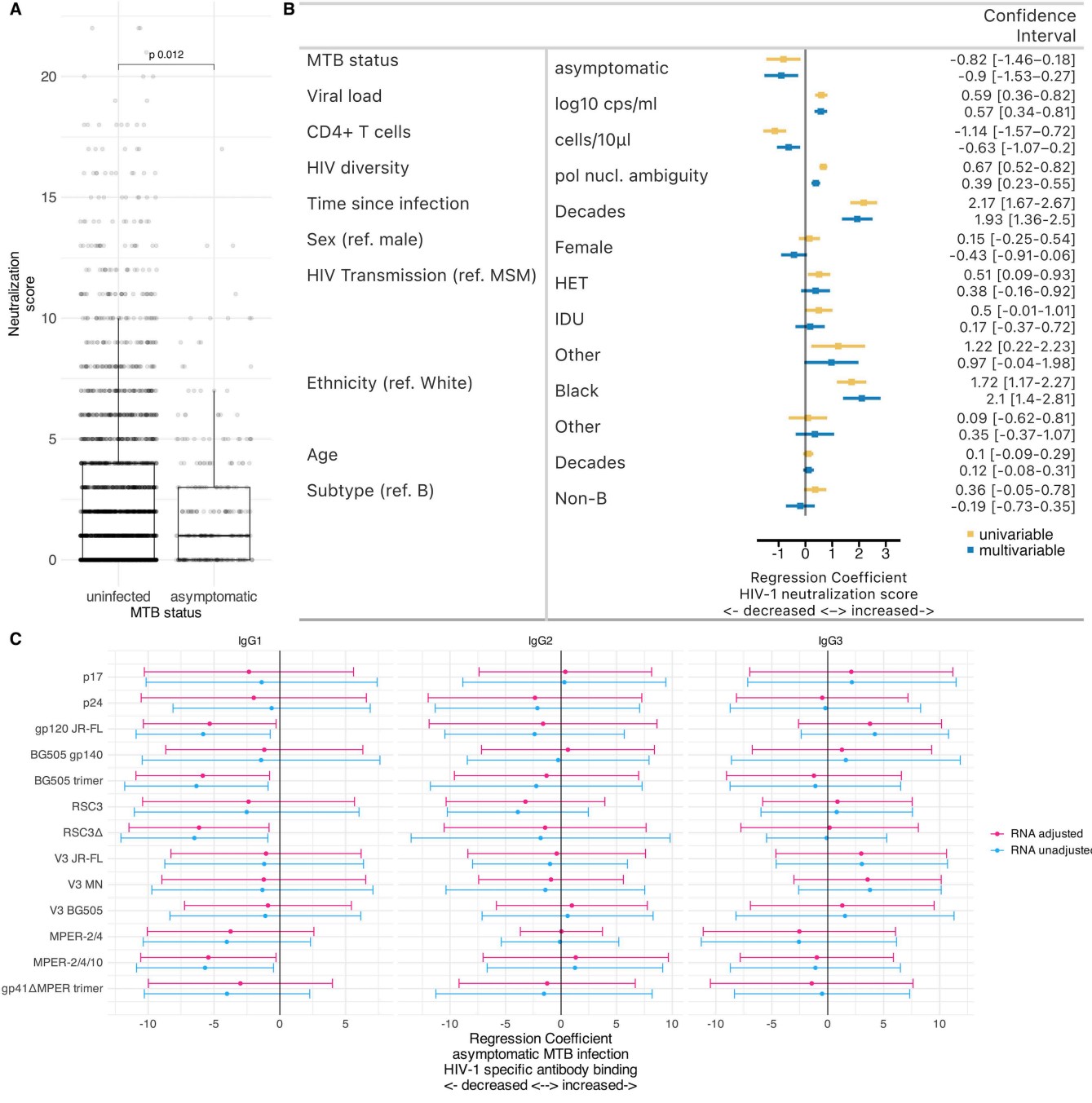

**Fig 2. People with HIV-1 and asymptomatic MTB infection exhibit reduced HIV-1 plasma neutralization and antigen binding.** A: Comparison of neutralization score at baseline between PWH with- and without asymptomatic MTB infection. Each box depicts the interquartile range (IQR), each vertical line extends to the most extreme value within 1.5 IQR from the box, and each horizontal line depicts the median. B: Adjusted comparison with demographic characteristics and HIV-1 disease specific parameters. The p value and effect estimates were determined with a tobit regression. C: Association between asymptomatic MTB infection and HIV-1 antigen plasma binding response, adjusted for demographic characteristics and HIV-1 disease specific parameters. RNA unadjusted corresponds to regression models not adjusted for HIV-1 RNA viral load. The effect estimates were determined with a linear regression and the confidence intervals were adjusted based on the p values adjusted for multiple testing using the Benjamini-Hochberg procedure.

of any trends (Fig 2C). The fact that IgG1 reactivity to BG505 trimer was reduced with asymptomatic MTB infection is intriguing and consistent with the reduced neutralization activity we observed, as IgG1 to BG505 trimer has previously been identified as a predictor of HIV-1 neutralization activity [43]. We observed no effects of asymptomatic MTB on HIV-1 Ab binding reactivity when restricting to PWH who were tested with an IGRA (S7 Fig). Overall, these results show that exposure to MTB has substantial associations with the formation of HIV-1 specific antibodies.

### Effects of other infections on HIV-1 plasma neutralization activity and on the HIV-1 binding antibody landscape

We determined if seropositivity to other infections exhibits similar patterns on the HIV-1 neutralization and HIV-1 Ab response. We assessed the association with serostatus of *Toxoplasma gondii* (n = 1374), CMV (n = 2360), and HCV (n = 623). The demographics are shown in S3 Table and the overlap between infections in S8 Fig. None of these pathogens reduced the HIV-1 neutralization like asymptomatic MTB, only CMV was associated with a 0.69 increase (CI = 0.11, 1.18) of neutralization (S6A Fig). Similarly, none of these pathogens were associated with reduced HIV-1 Ab reactivity (S9-S11 Figs). However, we observed increased IgG3 reactivity, against a range of antigens associated with HCV seropositivity (S11 Fig). Importantly, unlike MTB, seropositivity for none of these pathogens was associated with HIV-1 disease progression markers (HIV RNA levels or CD4 + T cell counts), underscoring the specificity of the TB-associated immune alterations (see S6C and S6D Fig).

### Shifts in HIV-1 antibody binding patterns demarcate progression to active TB

Finally, we assessed if Ab binding can be predictive of later active TB progression. While our analysis study population solely included individuals with asymptomatic MTB infection at baseline, longitudinal data available in the SHCS database allowed to identify 32 PWH that later progressed to active TB in a median time of 3.8 years (IQR 2.4 - 8.7), of which four had a prior asymptomatic MTB diagnosis. We thus next explored if the HIV-1 Ab binding in the asymptomatic MTB phase differ between active TB progressors and non-progressors. For this we categorized PWH in tertiles of combined Ab binding (excluding Gag antigens), respectively for IgG1, IgG2, and IgG3, and performed a time to event analysis. Intriguingly, PWH who later progressed to active TB had a significantly higher IgG3 (log rank test, p = 0.011, Fig 3A). This effect persisted in a cox proportional hazard regression model adjusted for demographic characteristics, where PWH in the third tertile of combined IgG3 Ab binding had a hazard ratio (HR) for active TB progression of 3.23 (CI = 1.16, 9.04) compared to PWH in the first tertile (S12 Fig). In contrast, PWH showed no differences in active TB progression across tertiles of IgG1 and IgG2 (Fig 3A).

For a better understanding of the association between IgG3 and active TB progression, we compared the individual IgG3 Ab binding responses between active TB progressors and -non-progressors (Fig 3B). This revealed significantly higher IgG3 Ab binding against V3 JR-FL, V3 MN, MPER-2/4, and MPER-2/4/10 (Fig 3B).

The increase in IgG3 responses among MTB infected PWH pointed towards a differential stimulation of HIV-1 Ab responses, as IgG3 responses are typically strongest in early HIV-1 infection [40]. This was particularly intriguing as active TB had been linked with a decrease in IgG3 [36]. In summary, these findings indicate that prior to the onset of active TB, PWH show a significant shift towards elevated IgG3 levels in their HIV-1 specific Abs.

In a sensitivity analysis assessing subsequent disease progression among all PWH, we identified similar associations between elevated HIV-specific IgG3 levels and the later development of bacterial pneumonia (HR 2.49, 95% CI: 1.32–4.66) and osteoporosis (HR 7.97, 95% CI: 1.63–38.91) (S13 Fig), suggesting that the IgG3 signature may reflect a broader shift in immune homeostasis.

## Discussion

In this study, we present substantial and consistent effects of mycobacterial exposure on the HIV-1 Ab response. Our findings indicate an association between asymptomatic MTB infection and reduced binding of HIV-1 antibodies to specific HIV-1 antigens, alongside diminished neutralization capacity.

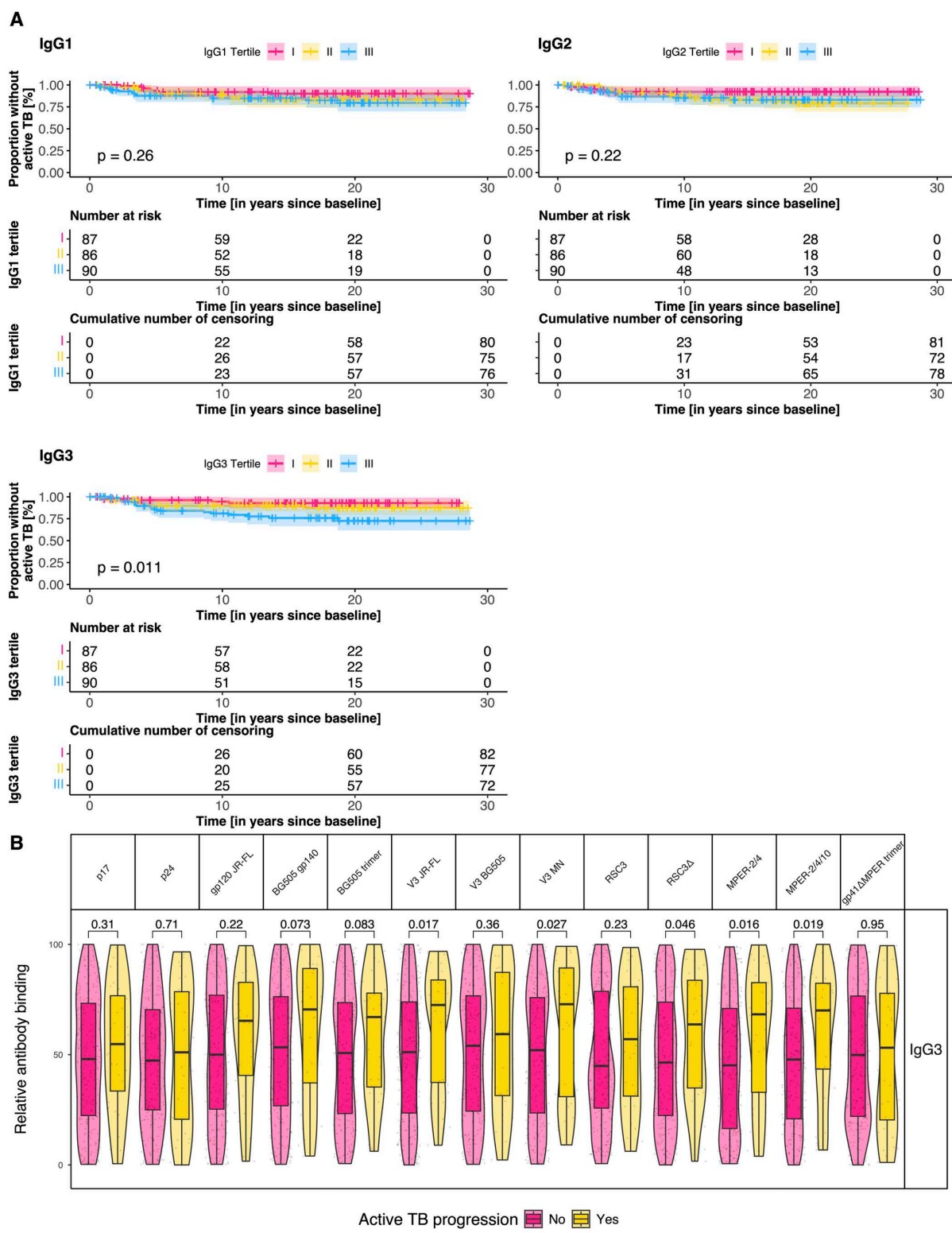

**Fig 3. Elevated HIV-1 specific IgG3 response associated with progression to active TB.** A. Kaplan-Meier curves of time to active TB progression from baseline, stratified by tertiles of combined HIV-1 specific IgG Ab class response. The p values estimates were determined with a log rank test. Cumulative number of censoring indicates the number people with loss to follow-up B: Comparison of the HIV-1 specific IgG3 Ab response against individual antigens between PWH with an asymptomatic MTB infection with (n=32) and without (n=231) later progression to active TB. P values were calculated with a two tailed Student's t test.

## Asymptomatic MTB infection and HIV-1 antibody response

Our findings demonstrate that asymptomatic MTB infection induces significant perturbations in HIV-1 Ab responses. These modulations are robust and persist even after adjusting for potential confounding factors. The observed down-regulation of IgG1 antibodies across multiple HIV-1 epitopes indicates a non-epitope-specific effect. If HIV-1 and MTB shared a common epitope, such as a glycan motif, we would expect a targeted antibody response rather than the observed broad modulation. This pattern suggests that the influence of MTB on HIV-1 Ab responses likely arises from a generalized immunological mechanism rather than shared antigenic features. One plausible mechanism is the alteration of the inflammatory microenvironment, potentially impacting antigen presentation and thereby contributing to the observed shifts in HIV-1 Ab responses. In support of this, prior transcriptomic analyses in PWH, with and without asymptomatic MTB co-infection, have shown innate immune gene alterations, aligning with this proposed mechanism [25]. Additionally, other research groups demonstrated that asymptomatic MTB infection elevates the frequency of NK cells in peripheral blood and perturbation in the myeloid compartment [27,51].

The proposed indirect effect of an asymptomatic MTB infection on HIV-1 immune responses aligns with established knowledge, as a lower viral load would naturally lead to reduced antigenic stimulation, thereby diminishing the HIV-1 Ab response. The observed direct effect of MTB infection on HIV-1 Ab responses, however, is more complex to interpret. One possible explanation for this direct influence is that chronic MTB infection may induce immune senescence or immune cell exhaustion, specifically impacting antigen-presenting or antibody-producing cells [52]. Such senescence could impair the immune system's ability to mount effective HIV-1-specific antibody responses.

We have extensively adjusted for demographic confounders, but as always in association studies like ours, the influence of as yet unidentified confounders cannot be excluded. Based on the data we have obtained and their agreement with previous observations, we are confident that the HIV-1 Ab shifts we observe are true signatures of an MTB-induced immune perturbation. It will be thus interesting in forthcoming studies to unravel the underlying mechanisms by which co-infection with MTB exhibit impacts HIV-1-specific Ab responses. While a reduction in viral load during asymptomatic MTB and the ensuing lowering of antigen, provides a plausible reason for lower binding and neutralizing antibody levels, the increase in particular IgG3 levels in PWH that develop acute TB, suggests a more complex impact on the HIV-1 antibody response. Overall, our findings are consistent with previous studies. Specifically, the previously demonstrated predictive role of the IgG1 BG505 trimer response in neutralization is confirmed by our observations, as we observe both a decrease in BG505 trimer response and a reduction in neutralization associated with MTB infection [43]. Additionally, the reduced HIV-1 RNA viral load aligns with our earlier studies investigating the impact of MTB infection on clinical outcomes in PWH, which identified a robust association between MTB infection and enhanced HIV-1 control, leading to decreased susceptibility to opportunistic infections in untreated PWH [21].

## Progression to active TB and IgG3 response

PWH with elevated anti-HIV-1 IgG3 levels were more likely to subsequently develop active TB, bacterial pneumonia, or osteoporosis, suggesting a broader humoral immune response. Notably, this IgG3 shift was detectable well before disease onset—with median lead times of 3.8 years for TB, and 8.3 and 8.1 years for pneumonia and osteoporosis, respectively. This is in contrast to the commonly observed HIV-1-specific immune trajectory, which is characterized by a peak of IgG3 in early infection and a shift to IgG1 as the disease progresses [40]. Of note, similar to our study Scriba et al., observed elevated total IgG in plasma of people with MTB infection without HIV infection who later progressed to active TB [7]. MTB specific Abs may not follow this pattern, as a recent reports indicates a decrease in MTB specific IgG3 Ab associated with active TB [36]. Similar, de Araujo et al. reported a protective effect of increased MTB specific IgG3 Abs against recurrent active TB [35]. In contrast, no difference in any IgG class (by bulk) was observed between PWH with active TB compared to PWH with asymptomatic MTB infection [48]. While another study confirms this effect in PWH, they

observed expanded IgG3 Ab with active TB compared to asymptomatic MTB infection in people without HIV [29]. From the other perspective, it was reported that SIV reduces MTB specific IgG responses in macaques [53].

In summary, our findings align with emerging evidence that MTB co-infection can modulate humoral immune responses in PWH [54]. Adeoye et al. [55] demonstrated that active MTB disease is associated with enhanced breadth and potency of HIV-1-specific antibodies, linked to specific plasma mediators involved in B cell development. Similarly, Nziza et al. [29] reported that while HIV infection compromises antibody responses, distinctions between latent and active TB based on antibody profiles remain detectable. Our findings and those of others suggest significant shifts in IgG subclass immune responses around the diagnosis of active TB, affecting both HIV-1 specific and MTB-specific antibodies. The shift towards IgG3 preceding active TB would need to be investigated in larger MTB-HIV cohorts to determine its potential as a novel diagnostic marker.

## Limitations

Establishing a causal relationship between MTB co-infection and the HIV-1 Ab response is challenging due to limited longitudinal data, particularly regarding the time of MTB infection. Additionally, our definition of MTB co-infection may include individuals with HIV who naturally cleared MTB infection, potentially leading to an underestimation of effect sizes. Furthermore, the small number of PWH progressing to active TB in our study population limits the statistical power of the IgG3 Ab response analysis. Moreover, non-progressors may progress at later timepoints outside the observation period.

Although B cell phenotyping and repertoire sequencing were not within the scope of the current study, our findings align with and extend previous work on B cell perturbations in HIV-1 infection. Liechti et al. [56] described significant alterations in the composition and phenotype of circulating B cell subsets in PWH, including an expansion of atypical memory and CD21⁻ naive B cells. These shifts reflect chronic immune activation and impaired B cell homeostasis. Our study builds on this foundation by demonstrating that MTB co-infection further modulates humoral immunity, particularly through changes in immunoglobulin isotype distribution, such as elevated IgG3 levels. This underscores the unique immunological footprint of MTB in the setting of HIV, warranting future efforts to link B cell phenotypes with antibody functionality and disease outcomes.

## Conclusion

Our study provides compelling evidence that MTB infection and progression to active TB significantly impact HIV-1 antibody development. PWH with asymptomatic MTB infection display reduced HIV-1 antibody levels and diminished neutralization capacity, possibly driven by MTB-induced changes in the innate immune system that reduce HIV-1 antigen levels and, consequently, HIV-1 viral load.

Moreover, individuals advancing to active TB exhibit a notable shift towards IgG3 in their HIV-1 Ab profile. This shift can be detected at least 180 days before the onset of active TB, with an average detection window exceeding four years prior to disease manifestation, indicating potential applications for early diagnostic markers. Finally, our results are consistent with the concept of contextual pathogenicity as described by Tutumlu et al. [57], where the immunological impact of a pathogen is influenced by the host's existing immune environment and co-infections. In this context, MTB co-infection appears to uniquely shape the humoral immune landscape in individuals with HIV, underscoring the importance of considering co-infections in immunological studies.

## Supporting information

**S1 Table. HIV-1 virus panel for neutralization assay** (originally published by Rusert et al. 2016, Nature Medicine, doi: https://doi.org/10.1038/nm.4187) [45].
(DOCX)

**S2 Table. HIV-1 antigens for binding assay** (originally published by Liechti et al. 2018, Journal of Immunological Methods, doi: https://doi.org/10.1016/j.jim.2017.12.003) [43].
(DOCX)

**S3 Table. Demographic characteristics by serostatus to various diseases.** Abbreviations: MTB (Mycobacterium tuberculosis), CMV (Cytomegalovirus), Toxoplasma (Toxoplasma gondii), and HCV (Hepatitis C virus).
(DOCX)

**S1 Fig. Plasma neutralization of eight HIV-1 strains.** Distributions of plasma neutralization ranging from 0 to 100, whereas 100 indicates full neutralization of the respective HIV-1 strain. MTB status was defined as i) MTB uninfected, defined through presence of a negative tuberculin skin test (TST) or interferon gamma release assay (IGRA) any time before or 1 year post baseline. ii) Asymptomatic MTB infection, defined either through a positive TST or IGRA any time before or 1 year post baseline, or as progression to active TB > 180 days post baseline P values are derived from a tobit model additional adjusted for demographic characteristics and HIV-1 disease specific parameters.
(TIF)

**S2 Fig. Plasma Ab binding of 13 HIV-1 antigens in dependence of MTB status stratified by IgG class.** Ab binding was measured as mean fluorescent intensity (MFI) transformed to a relative binding ranging from 0 to 100, whereas 100 indicates strongest binding. MTB status was defined as i) MTB uninfected, defined through presence of a negative tuberculin skin test (TST) or interferon gamma release assay (IGRA) any time before or 1 year post baseline. ii) Asymptomatic MTB infection, defined either through a positive TST or IGRA any time before or 1 year post baseline, or as progression to active TB > 180 days post baseline.
(TIF)

**S3 Fig. HIV-1 neutralization score and associations with demographic characteristics and HIV-1 disease specific parameters.** Neutralization score was determined from observed plasma neutralization against eight HIV-1 strains from five subtypes, ranging from 0 to 100% neutralization. For each strain a score was calculated, 3 when neutralization >80%, 2 when neutralization 50% to <80%, 1 when neutralization 20% to <50%, and 0 when neutralization <20%. The sum of the individual virus scores were then summarized as the neutralization score (ranging from 0 to maximum 24). The effect estimates were determined with a tobit regression.
(TIF)

**S4 Fig. HIV-1 specific IgG1/2/3 antigen binding and associations with demographic characteristics and HIV-1 disease specific parameters.** Ab binding was measured as mean fluorescent intensity (MFI) transformed to a relative binding ranging from 0 to 100, whereas 100 indicates strongest binding. Inner circles describe univariable results and brackets describe multivariable results, adjusted for all other demographic characteristics and HIV-1 disease specific parameters. The p values and effect estimates were determined with a linear regression.
(TIF)

**S5 Fig. Asymptomatic MTB infection and associations with demographic characteristics and HIV-1 disease specific parameters.** MTB status was defined as i) MTB uninfected, defined through presence of a negative tuberculin skin test (TST) or interferon gamma release assay (IGRA) any time before or 1 year post baseline. ii) Asymptomatic MTB infection, defined either through a positive TST or IGRA any time before or 1 year post baseline, or as progression to active TB > 180 days post baseline. The effect estimates were determined with a logistic regression.
(TIF)

**S6 Fig. Infection Serostatus & HIV Response. A:** HIV-1 neutralization score and associations with demographic characteristics and HIV-1 disease specific parameters. The effect estimates were determined with a tobit regression.

**B/C:** Serostatus of MTB, Toxoplasma gondii, CMV, and HCV and associations with HIV RNA load and CD4 + T cell counts. MTB serostatus status was defined as i) MTB uninfected, defined through presence of a negative tuberculin skin test (TST) or interferon gamma release assay (IGRA) any time before or 1 year post baseline. ii) Asymptomatic MTB infection, defined either through a positive TST or IGRA any time before or 1 year post baseline, or as progression to active TB > 180 days post baseline. CMV, Toxoplasma gondii, and HCV serostatus were defined based on IgG antibody tests any time before or 1 year post baseline. The effect estimates were determined with a logistic regression. (TIF)

**S7 Fig. Association between asymptomatic MTB infection (determined by interferon gamma release assay (IGRA) and HIV-1 antigen plasma binding response.** Adjusted for demographic characteristics and HIV-1 disease specific parameters. RNA unadjusted corresponds to regression models not adjusted for HIV-1 RNA viral load. The effect estimates were determined with a linear regression and the confidence intervals were adjusted based on the p values adjusted for multiple testing using the Benjamini-Hochberg procedure. (TIF)

**S8 Fig. Venn diagram of Serostatus.** Depicted is the overlap of seropositive tests in the study population against MTB (Mycobacterium tuberculosis), CMV (Cytomegalovirus), TOXO (Toxoplasma gondii), and HCV (Hepatitis C virus). (TIF)

**S9 Fig. Association between CMV serostatus and HIV-1 antigen plasma binding response.** Adjusted for demographic characteristics and HIV-1 disease specific parameters. RNA unadjusted corresponds to regression models not adjusted for HIV-1 RNA viral load. The effect estimates were determined with a linear regression and the confidence intervals were adjusted based on the p values adjusted for multiple testing using the Benjamini-Hochberg procedure. (TIF)

**S10 Fig. Association between Toxoplasma gondii serostatus and HIV-1 antigen plasma binding response.** Adjusted for demographic characteristics and HIV-1 disease specific parameters. RNA unadjusted corresponds to regression models not adjusted for HIV-1 RNA viral load. The effect estimates were determined with a linear regression and the confidence intervals were adjusted based on the p values adjusted for multiple testing using the Benjamini-Hochberg procedure. (TIF)

**S11 Fig. Association between HCV serostatus and HIV-1 antigen plasma binding response.** Adjusted for demographic characteristics and HIV-1 disease specific parameters. RNA unadjusted corresponds to regression models not adjusted for HIV-1 RNA viral load. The effect estimates were determined with a linear regression and the confidence intervals were adjusted based on the p values adjusted for multiple testing using the Benjamini-Hochberg procedure. (TIF)

**S12 Fig. Time until active TB since date of antibody response measurement adjusted for mean tertiles of IgG3 antigen binding.** Adjusted comparison with demographic characteristics and HIV-1 disease specific parameters. Progression to active TB was defined as diagnosis of active TB > 180 days post baseline. Comparison group was defined through a positive TST or IGRA any time before or 1 year post baseline. The effect estimates were determined with a cox proportional hazard regression. (TIF)

**S13 Fig. Time until opportunistic infection or non-communicable disease since date of antibody response measurement depended on overall IgG3 antigen binding.** Adjusted comparison with demographic characteristics and HIV-1 disease specific parameters. Progression to opportunistic infection or non-communicable disease was defined as

diagnosis of the respective disease >180 days post baseline. The effect estimates were determined with a cox proportional hazard regression.
(TIF)

## Acknowledgments

The authors thank the patients who participated in the Swiss HIV Cohort Study; the physicians and study nurses for the excellent patient care provided to participants; Katja Benic and Sandra E. Chaudron from the Swiss HIV Cohort Study data center for data management; and Danièle Perraudin and Marianne Amstad for administration.

The SHCS data are gathered by the Five Swiss University Hospitals, two Cantonal Hospitals, 15 affiliated hospitals and 36 private physicians (listed in http://www.shcs.ch/180-health-care-providers).

## Author contributions

**Conceptualization:** Marius Zeeb, Katharina Kusejko, Huldrych F. Günthard, Roger D. Kouyos, Alexandra Trkola, Johannes Nemeth.

**Data curation:** Marius Zeeb, Peter Rusert, Thomas Liechti, Claus Kadelka, Huldrych F. Günthard, Roger D. Kouyos, Alexandra Trkola, Johannes Nemeth.

**Formal analysis:** Marius Zeeb.

**Funding acquisition:** Marius Zeeb, Huldrych F. Günthard, Roger D. Kouyos, Alexandra Trkola, Johannes Nemeth.

**Investigation:** Marius Zeeb, Huldrych F. Günthard, Roger D. Kouyos, Alexandra Trkola, Johannes Nemeth.

**Methodology:** Marius Zeeb, Sonja Hartnack, Huldrych F. Günthard, Roger D. Kouyos, Alexandra Trkola, Johannes Nemeth.

**Project administration:** Marius Zeeb, Johannes Nemeth.

**Supervision:** Huldrych F. Günthard, Roger D. Kouyos, Alexandra Trkola, Johannes Nemeth.

**Validation:** Huldrych F. Günthard, Roger D. Kouyos, Alexandra Trkola, Johannes Nemeth.

**Visualization:** Marius Zeeb.

**Writing – original draft:** Marius Zeeb, Huldrych F. Günthard, Roger D. Kouyos, Alexandra Trkola, Johannes Nemeth.

**Writing – review & editing:** Marius Zeeb, Katharina Kusejko, Sonja Hartnack, Chloé Pasin, Irene A. Abela, Peter Rusert, Thomas Liechti, Claus Kadelka, Julia Notter, Anna Eichenberger, Matthias Hoffmann, Hans H. Hirsch, Alexandra Calmy, Matthias Cavassini, Niklaus D. Labhardt, Enos Bernasconi, Huldrych F. Günthard, Roger D. Kouyos, Alexandra Trkola, Johannes Nemeth.

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
