## [Decision Letter · Decision Letter 0]

9 Apr 2025

Infection with Mycobacterium tuberculosis alters the antibody response to HIV-1

PLOS Pathogens

Dear Dr. Zeeb,

Thank you for submitting your manuscript to PLOS Pathogens. After careful consideration, we feel that it has merit but does not fully meet PLOS Pathogens's publication criteria as it currently stands. Therefore, we invite you to submit a revised version of the manuscript that addresses the points raised during the review process.

Please submit your revised manuscript within 60 days Jun 07 2025 11:59PM. If you will need more time than this to complete your revisions, please reply to this message or contact the journal office at plospathogens@plos.org. Please include the following items when submitting your revised manuscript:

We look forward to receiving your revised manuscript.

Kind regards,

Susan R. Ross, PhD

Section Editor

PLOS Pathogens

Editor-in-Chief

PLOS Pathogens

Editor-in-Chief

PLOS Pathogens

orcid.org/0000-0002-7699-2064

**Journal Requirements:**

1) Please provide an Author Summary. This should appear in your manuscript between the Abstract (if applicable) and the Introduction, and should be 150-200 words long. The aim should be to make your findings accessible to a wide audience that includes both scientists and non-scientists. Sample summaries can be found on our website under Submission Guidelines:

https://journals.plos.org/plospathogens/s/submission-guidelines#loc-parts-of-a-submission

3) We notice that your supplementary Figures, and Tables are included in the manuscript file. Please remove them and upload them with the file type 'Supporting Information'. Please ensure that each Supporting Information file has a legend listed in the manuscript after the references list.

Potential Copyright Issues:

i) Figure 1. Please confirm whether you drew the images / clip-art within the figure panels by hand. If you did not draw the images, please provide (a) a link to the source of the images or icons and their license / terms of use; or (b) written permission from the copyright holder to publish the images or icons under our CC BY 4.0 license. Alternatively, you may replace the images with open source alternatives. See these open source resources you may use to replace images / clip-art:

ii) Supplementary Table 1 : Thank you for stating "originally published by Rusert et al. 2016 Nature Medicine." The source of the table is not compatible with our open access license. Please remove / replace the table. 

5) Regarding your Data Availability Statement, please provide more details on how readers can access the data ,e.g. an email address or web link that can be used to request data access.

**Reviewers' Comments:**

Reviewer's Responses to Questions

**Part I - Summary**

Reviewer #1: This article focuses on the interaction between TB/HIV co-infection and the specific B cell response to HIV. This retrospective study analysed 263 HIV-infected individuals (231 with latent tuberculosis and 32 with active tuberculosis during follow-up). The authors tested the levels of HIV antibodies in the two groups and then compared them with those infected with HIV without TB infection. The main conclusion is that active or asymptomatic TB infection alters the development of HIV-specific antibodies.

Strengths

- The study includes a well-established, well-followed cohort

- The tests described appear to be standardised in the laboratory.

Weaknesses

- Selection of individuals to be analysed

- Control groups to validate the analysis

Reviewer #2: This brief manuscript from Zeeb, et al. describes a retrospective study of plasma from PWH with or without Mtb infection to investigate antibodies binding/neutralization to HIV. The data suggests that Mtb infection, whether subclinical or symptomatic, can influence antibody responses to HIV. While many studies have looked at Ab responses to HIV or to Mtb, few have looked at the effect of TB status on anti-HIV responses. Published data from the SHCS offered a unique opportunity to do so. Importantly, only data from PWH off ART were included. Appropriate statistical tests were applied. The analyses revealed that Mtb coinfection was associated with reduced HIV-1 neutralization. Interestingly, PWH who went on develop active TB exhibited higher IgG3 even though prior studies suggest that IgG3 protects from TB disease. This inconsistency is adequately discussed and the limitations of this analysis are stated.

In summary, this retrospective study of the influence of Mtb on anti-HIV responses is novel. It is also very simplistic as Ab glycosylation or functional roles are not considered, presumably because these data do not exist from the sample set. The report is well-written, the analyses are sufficient for the stated purpose, and the conclusions are not over-drawn. However, the limited scope of the serological data likewise limits the significance of this contribution.

Reviewer #3: This manuscript by Zeeb et al builds on a previous publication from this group showing that in the Swiss HIV Cohort study, showing a surprising association between LTBI or Mtb-uninfected status and lower HIV set point viral load (SPVL) when compared to PWH with Active TB. They also reported fewer opportunistic infections overall in HIV/Mtb coinfected patients with LTBI. Here they attempt to investigate (in a retrospective study) the potential immunological underpinnings of the association that they previously reported.

Using the SHCS, they studied PWH with and without TST+/IGRA+ tests at baseline and assessed HIV-1-specific plasma binding/neutralizing responses in individuals who remained asymptomatic and in those who developed active TB. They report that PWH with asymptomatic Mtb had reduced HIV-1 Ab levels and neutralization capacities compared to PWH with no Mtb while PWH who developed active TB showed a preferential shift to IgG3 Abs, showing that PWH in these 2 categories diverge in their Ab responses to HIV.

Overall the findings are interesting and the study is well designed. Combined with the results from their previous study in PLos Biology, their observations raise some interesting issues regarding how prior HIV infection might potentially influence Mtb infection outcomes. However, while these findings report an association between asymptomatic MTB infection and reduced binding of HIV-1 Abs antibodies to specific HIV antigens and, reduced neutralization capacity, the underlying correlates and mechanisms remain unclear.

**Part II – Major Issues: Key Experiments Required for Acceptance**

Reviewer #1: General comments

In terms of B cell responses, the interaction between TB infection and HIV is very complex. Both monoinfections induce changes in circulating B cells, B cell repertoire and levels of specific and non-specific immunoglobulins. Both TB and HIV induce polyclonal activation, and this polyclonal activation can alter the levels of other specific antibodies. The changes in circulating B cells and B cell repertoire are described in latent and active tuberculosis. IgG3 (specific or non-specific) are not only described in tuberculosis but in other intracellular infections. Thus, the question is whether the results described here are specific to tuberculosis infection or observed in other intracellular co-infections. Furthermore, are these only changes induced by tuberculosis infection? Thus, results presented here do not allow us to distinguish between tuberculosis mono-infection and HIV infection.

Reviewer #2: None

Reviewer #3: Admittedly, as acknowledged by the authors, establishing causal links is challenging, however since longitudinal samples appear to be available through the SHCS, there is potential to generate these data to strengthen the study. Other data that could strengthen the study is to provide evidence of immune perturbations by analyzing plasma for inflammatory mediators that might correlate with outcomes.

While IGRA+/TST+ data are reported the numbers who were IGRA+ is not outlined. Since TST does not distinguish between BCG and Mtb, it is important to show analysis of the IGRA+ results separately in order to interpret the impact of asymptomatic Mtb infection.

Overall, the data presented here are intriguing but fall short in the absence of providing additional data to provide insight into the biological implications behind the observations.

**Part III – Minor Issues: Editorial and Data Presentation Modifications**

Reviewer #1: Specific comments

- Some controls are missing in my opinion (e.g. TB monoinfection or other coinfections) to know the changes in B cell response associated to.

- Are other than HIV specific B cell responses altered in the same way in TB-HIV co-infection?

- Reference is required in line 82

- Some biological measures will be described, such as immunoglobulin levels and isotype distribution.

- Circulating B cell levels and B cell repertoire must be described.

- In Figure 2, it will be better to write viral load instead of RNA.

- Better revision of published articles in the field.

Reviewer #2: None

Reviewer #3: (No Response)

PLOS authors have the option to publish the peer review history of their article (what does this mean? ). If published, this will include your full peer review and any attached files.

**Do you want your identity to be public for this peer review?** For information about this choice, including consent withdrawal, please see our Privacy Policy .

Reviewer #1: **Yes: ** Daniel SCOTT

Reviewer #2: No

Reviewer #3: No

**Figure resubmission:**

**Reproducibility:**



---

## [Decision Letter · Decision Letter 1]

7 Jul 2025

Dear Mr. Zeeb,

We are pleased to inform you that your manuscript 'Infection with Mycobacterium tuberculosis alters the antibody response to HIV-1' has been provisionally accepted for publication in PLOS Pathogens.

Best regards,

Guido Silvestri

Academic Editor

PLOS Pathogens

Susan Ross

Section Editor

PLOS Pathogens

Sumita Bhaduri-McIntosh

Editor-in-Chief

PLOS Pathogens

orcid.org/0000-0003-2946-9497

Michael Malim

Editor-in-Chief

PLOS Pathogens

orcid.org/0000-0002-7699-2064

Reviewer Comments (if any, and for reference):

Reviewer's Responses to Questions

**Part I - Summary**

Reviewer #1: Dear author,

Thank you for your answers.

I think communication has improved with the new data.

However, the article is still very descriptive, and the mechanisms by which tuberculosis interferes with the HIV antibody are still unclear. I hope you can continue the research in this area.

Best wishes

Reviewer #2: This revised manuscript by Zeeb, et al. retains all it’s original strengths. This retrospective serologic study demonstrates that Mtb co-infection can influence antibody responses to HIV. The authors further strengthened this conclusion with additional data showing that antibodies to several other opportunistic infections are NOT altered by Mtb. The report is well-written, the analyses are sufficient for the stated purpose, and the conclusions are well-supported. While the limited scope of the assays (e.g. Ab glycosylation and/or function was not assessed) remains valid, it does not detract substantially from the novel aspects of the work presented.

Reviewer #3: authors have addressed most of the concerns and the limitations they describe are noted

**Part II – Major Issues: Key Experiments Required for Acceptance**

Reviewer #1: Due to the limitations of this retrospective study, there is no need for further experiments.

Reviewer #2: None

Reviewer #3: none

**Part III – Minor Issues: Editorial and Data Presentation Modifications**

Reviewer #1: There are no minor issues.

Reviewer #2: None

Reviewer #3: none

PLOS authors have the option to publish the peer review history of their article (what does this mean? ). If published, this will include your full peer review and any attached files.

**Do you want your identity to be public for this peer review?** For information about this choice, including consent withdrawal, please see our Privacy Policy .

Reviewer #1: No

Reviewer #2: No

Reviewer #3: No

---

## [Editor Report · Acceptance letter]

Dear Mr. Zeeb,

We are delighted to inform you that your manuscript, " 

Infection with Mycobacterium tuberculosis alters the antibody response to HIV-1," has been formally accepted for publication in PLOS Pathogens.

Best regards,

Sumita Bhaduri-McIntosh

Editor-in-Chief

PLOS Pathogens

orcid.org/0000-0003-2946-9497

Michael Malim

Editor-in-Chief

PLOS Pathogens

orcid.org/0000-0002-7699-2064